# Changes in HRSV Epidemiology but Not Circulating Variants in Hospitalized Children due to the Emergence of SARS-CoV-2

**DOI:** 10.3390/v15061218

**Published:** 2023-05-23

**Authors:** Monika Jevšnik Virant, Manca Luštrek, Rok Kogoj, Miroslav Petrovec, Tina Uršič

**Affiliations:** Institute of Microbiology and Immunology, Faculty of Medicine, University of Ljubljana, Zaloška 4, 1000 Ljubljana, Slovenia; manca.lustrek@mf.uni-lj.si (M.L.); rok.kogoj@mf.uni-lj.si (R.K.); mirc.petrovec@mf.uni-lj.si (M.P.); tina.ursic@mf.uni-lj.si (T.U.)

**Keywords:** human respiratory syncytial virus, SARS-CoV-2, epidemiology, genotyping

## Abstract

This study assesses the circulation of human respiratory syncytial virus (HRSV) genotypes before, during, and toward the end of the severe acute respiratory syndrome coronavirus 2 (SARS-CoV-2) pandemic in children and determines the influence of the pandemic on HRSV circulation patterns and evolution. Phylogenetic analysis of the hypervariable glycoprotein G gene was performed on 221/261 (84.7%) HRSV-positive samples and shows two separated clusters, one belonging to HRSV-A (129/221) and another to HRSV-B (92/221). All Slovenian HRSV-A strains contained the 72-nucleotide-long duplicated region in the attachment glycoprotein G gene and were classified as lineage GA2.3.5. All Slovenian HRSV-B strains similarly contained a 60-nucleotide-long duplicated region in the attachment glycoprotein G gene and were classified as lineage GB5.0.5a. During the 3-year period (2018–2021) covered by the study, no significant differences were observed within strains detected before the SARS-CoV-2 pandemic, during it, and after the implementation of nonpharmaceutical preventive measures. Slovenian HRSV-A strains seem to be more diverse than HRSV-B strains. Therefore, further whole-genome investigations would be required for better monitoring of the long-term impact of SARS-CoV-2 endemic circulation and the formation of new HRSV lineages and epidemiological patterns.

## 1. Introduction

Human respiratory syncytial virus (HRSV) is the major cause of respiratory infections worldwide in infants and young children. Human RSV is an enveloped, single-stranded, negative-sense RNA virus belonging to the *Pneumoviridae* family. Strains can be genetically divided into two subgroups, HRSV-A and HRSV-B, and further into multiple genotypes that can be linked to differences in disease severity [1,2]. Currently, 13 HRSV genotypes have been defined within subgroup A (GA 1-7, SAA1, NA1-4, and ON1-2) and 20 genotypes within subgroup B (GB 1-4, SAB1-4, URU1-2, and BA1-10). Goya et al. recently suggested new rules for HRSV classification that decrease the number of genotypes to three for HRSV-A (GA1–GA3) and seven for HRSV-B (GB1–GB7) [3]. Currently, the main circulating genotype within HRSV-A is ON1, which was first described in 2010 and has since become widespread globally. It is defined by a 72-nucleotide (nt) duplication in the hypervariable region of the protein G gene that is not present in other genotypes. Similarly, in genotype BA strains within subgroup HRSV-B, a duplication of 60 nt in the gene G was also found in 1999 [4,5].

It has been widely accepted that cold temperatures stabilize the HRSV lipid envelope, which causes HRSV to peak in regular annual winter epidemics. Due to high humidity, rainy periods, and low outside temperatures, people mostly stay indoors, allowing HRSV to be transmitted more easily [6,7]. In a longitudinal study from Slovenia, which included respiratory samples from hospitalized patients during an 8-year period before the emergence of SARS-CoV-2 (2012 to 2020), HRSV displayed a regular pattern, mostly circulating in the winter months from October to April with a peak in January. After March 2020, when SARS-CoV-2 was first detected in the Slovenian population, HRSV did not display the usual epidemiological profile, and almost no cases were detected in the population for roughly 1 year. Interestingly, after the Slovenian government started gradually lifting SARS-CoV-2 nonpharmaceutical preventive measures, an unusually high number of HRSV cases started to appear in the middle of summer, despite high outside temperatures of around 30 °C [8].

Before SARS-CoV-2, in most European countries multiple genetically distinct lineages were simultaneously circulating at the same time in the same location or widely separated geographic locations. Every season, different strains, genotypes, or clades usually cocirculate, with the predominance of one over the others. Such a circulation pattern and predominant genotype replacement with another between years were readily observed [9,10]. Considering the 2 years of the SARS-CoV-2 epidemic and the complete disappearance of HRVS in one annual season, it would be of great interest to obtain insight into the dynamics of HRSV genotypes before, during, and after the emergence of SARS-CoV-2, especially due to an unusual epidemiological pattern observed after the alleviation of SARS-CoV-2 preventive measures. Moreover, these data could help us predict HRSV genotypes that will circulate in the following seasons. Therefore, this study assesses the circulation of HRSV genotypes before and after the emergence of SARS-CoV-2 to determine the influence of the SARS-CoV-2 pandemic on HRSV circulation patterns and evolution.

## 2. Materials and Methods

### 2.1. Study Population

From January 2018 to December 2021, a total of 125,072 nasopharyngeal swabs (NPs) from pediatric patients (aged 0 to 18 years) were sent to the Institute of Microbiology and Immunology, Faculty of Medicine, University of Ljubljana, for routine molecular detection of respiratory viruses. From these, 6423 NPs originated from hospitalized children at the Pediatric Unit of the Ljubljana University Medical Center, thus excluding the 118,649 children tested only for SARS-CoV-2 since these include also the screening of asymptomatic individuals. From all 6423 NPs of hospitalized children, 789 were HRSV-positive (from 6 days to 10 years old) and 261/789 HRSV samples contained suitable viral loads (Ct < 27) to be included in further analysis (Figure 1). After sequencing, 129 HRSV-A and 92 HRSV-B samples displayed sequences of suitable length and quality for phylogenetic analysis. The National Medical Ethics Committee of the Republic of Slovenia approved the study protocol on 18 September 2020 (no. 0120-349/2020-6).

### 2.2. Sample Preparation, Nucleic Acid Extraction, and Diagnostic Real-Time RT-PCR

NP swabs were collected using flocked-tip swabs and transported to the laboratory in a Copan universal transport medium UTM-RT (Copan Italia, Brescia, Italy) for routine diagnostics. Total nucleic acids were isolated from 200 μL of vigorously vortexed NP swab medium using Total Nucleic Acid Isolation Kit I (RocheApplied Science, Mannheim, Germany) on a MagNa Pure Compact instrument (RocheApplied Science, Mannheim, Germany), according to the manufacturer’s instructions. HRSV and all other respiratory viruses, including human coronaviruses (HCoVs), human rhinoviruses (HRVs), human metapneumovirus (HMPV), human bocavirus (HBoV), human mastadenoviruses (HAdVs), parainfluenza viruses 1–4 (PIV 1–4), enteroviruses (EVs), human parechovirus (HPeV), and influenza viruses A and B (Flu-A and -B) were detected by using Respiratory Viruses 16-Well Assay V.17 (AusDiagnostics, Mascot, Australia) until 2020. After validation of the Respiratory Viruses 16-Well Assay V.19 in 2020, which also detects SARS-CoV-2, this version was used until the end of the study period [11]. After testing, isolated NAs were immediately stored at −20 °C.

### 2.3. HRSV Hypervariable G Gene Region Nested RT-PCR and Sanger Sequencing

Based on the results of the Respiratory Viruses 16-Well Assay V.17 and V.19, the remainder of 261 HRSV-positive stored NA samples were used to perform sequence analysis. Nested PCR was carried out with two sets of primers (0.4 µM of each primer in both reactions) and 1 µL of NA to amplify the second hypervariable region (HVR2) of the HRSV G gene using the primers and protocol described by Slovic et al. [12]. The nested PCR was performed by using the PrimeScript One Step RT-PCR kit (Takara Bio Inc., Shiga, Japan) in a Veriti™ 96-Well Fast Thermal Cycler (Thermo Fisher Scientific, Waltham, MA, USA) following the thermal cycling conditions as described previously [12]. The amplified products (501–573 bp for HRSV subgroup A and 560–564 bp for HRSV subgroup B strains) were detected by 1.5% agarose gel electrophoresis and sequenced by Sanger sequencing. Sequencing reactions were set up with Fast AP + Exonuclease I (Thermo Fischer Scientific, Waltham, MA, USA) enzymatically purified second-stage PCR products, using the primers as described by Slovic et al. [12] and BigDye Terminator v3.1 Cycle Sequencing Kit (Thermo Fisher Scientific, Waltham, MA, USA) according to the manufacturer’s instructions. Sequencing was performed on an ABI-3500 Genetic Analyser (Thermo Fisher Scientific, Waltham, MA, USA).

### 2.4. Data Analysis

Nucleotide sequences were compared with previously described HRSV G gene sequences from the GenBank database, using the Basic Local Alignment Search Tool (BLAST) available at the NCBI website (https://blast.ncbi.nlm.nih.gov/Blast.cgi?PROGRAM=blastn&PAGE_TYPE=BlastSearch&LINK_LOC=blasthome, accessed on 6 March 2023) for basic differentiation into HRSV subgroup A and subgroup B. In addition, 357 bp-long sequences were aligned using the ClustalW algorithm using Molecular Evolutionary Genetics Analyses (MEGA-X) software v.10.1.7 along with reference sequences (GenBank acc. nos. JN257693, AY343558, AY333364, and KP258745) with and without duplications in the G gene of HRSV-A and -B. Mutations in the original and duplicated regions were further analyzed at the nucleotide and amino acid levels. Genotyping and phylogenetic analyses were performed using Nexclade software v2.14.1 [13], which is based on the genotyping scheme proposed by Goya et al. [3].

## 3. Results

From a total of 6423 tested NPs from children, 789/6423 (12.3%) were detected as HRSV positive in routine diagnostics. Children tested for respiratory viruses had a median age of 63.7 months, IQR 10.1–73.8, with no differences between the years included in the study. Children positive for HRSV had a lower median age compared to the entire pediatric population; 29.3 months, IQR 3.3–32.6. Similarly, no difference was observed between the years included in the study. Based on the Ct value (Ct value below 27), 261 HRSV-positive NP samples of pediatric patients were included in further analysis. The median age of this study participant subpopulation was 25.8 months, IQR 2.4–28.2 with a female: male ratio of 1:1.3 (114/261; 43.7% females). Table 1 shows the yearly distribution of pediatric nasopharyngeal samples and HRSV genotyping results.

In 2018 and 2019, the seasonality of HRSV cases remained as seen before the SARS-CoV-2 pandemic. After the appearance of the first positive case of SARS-CoV-2 in Slovenia on 4 March 2020, the number of HRSV-positive cases began to decline rapidly. Between July 2020 and May 2021, no cases of HRSV were detected at all (Figure 2). After a 1-year period without the usual HRSV epidemic pattern, positive cases started to reappear and very quickly increased in number during late spring and summer 2021 after alleviating SARS-CoV-2 preventive/protective measures.

From the 261 samples included in the sequencing part of the study, BLAST analysis revealed a total of 144/261 (55.2%) belonging to group HRSV-A and 117/261 (44.8%) to group HRSV-B. A temporal analysis showed that before the emergence of SARS-CoV-2, the predominant subgroup was HRSV-B (91/132, 68.9%), whereas, after a 1-year absence of cases, group A caused an unusual epidemic during the late summer/autumn surge of cases (103/129, 79.8%; Figure 3). In the interim period, when SARS-CoV-2 preventive measures were fully in place, we observed a decrease in the number of samples submitted for respiratory virus diagnostics in comparison to the previous and following periods and a rise in the number of samples submitted only for SARS-CoV-2 diagnostics (54,954 samples from March 2020 to March 2021 and 63,696 samples from April 2021 to December 2021).

A further genetic analysis was performed on 221/261 (84.7%) sequenced samples (129 HRSV-A and 92 HRSV-B) that had suitable sequence quality. Samples with a 357-bp fragment for HRSV-A and 362-bp fragment for HRSV-B spanning the hypervariable attachment glycoprotein G gene region were aligned with MEGA-X, including available reference sequences from the GenBank database. The phylogenetic analysis confirmed the results from BLAST with no discrepancies and showed a classification of samples into two clearly separated clusters; namely, subgroup HRSV-A (*n* = 129) and HRSV-B (*n* = 92).

All 129 Slovenian HRSV-A sequences contained the 72-nucleotide-long duplicated region at the end of the G gene and were closely related to the original Canadian ON1 genotype (GeneBank acc. no. JN257693) first detected in 2010, and all of them, regardless of the year of collection, were further classified by Nextclade as lineage GA2.3.5 (Table 1 and Figure 4).

Similarly, all 92 Slovenian HRSV-B contained the 60-nucleotide-long duplicated region in the G gene and were closely related to the English BA strain (GeneBank acc. no. KY249660), and all of them, regardless of the year of collection, were further classified by Nextclade as lineage GB5.0.5a (Table 1 and Figure 5).

The amino acid analysis of the hypervariable attachment protein of the HRSV-A subgroup—the original region (amino acids 261–283) and repeated region (amino acids 285–307) compared with the original Canadian ON1 genotype (GeneBank acc. no. JN257693)—is shown in detail in Figure 6 and Table 2. Numerous amino acid changes were observed with variable frequencies over the investigating period independently in both regions (the original and the repeated regions). In greater detail, we compared the highly conserved sequence motif (GYLSPSQ) as seen in the original Canadian ON1 genotype (GeneBank acc. no. JN257693), where the exact-same motif was duplicated (amino acids 272–278 and 296–302) with the 129 Slovenian HRSV-A strains. Only in three Slovenian sequences did the motif GYLSPSQ remain the same in both regions. In 26/129 (20.2%) sequences, one mutation was found in the first region and two mutations in the second region (motif GYPSPSQ in the duplicated region to motif GHPSPSQ in the repeated region). On the other hand, in 13/129 (10.1%) Slovenian sequences, two mutations were found in the originating region (motif GYPSQSQ) and only one mutation in the repeated region (motif GYPSPSQ). All other motifs in both regions occurred at lower frequencies (Figure 6).

The amino acid analysis of the hypervariable region of the attachment G protein of the HRSV-B subgroup—the original region (amino acids 240–259) and the repeated region (amino acids 260–279) compared with the Madrid BA genotype (GenBank acc. no. AY333364)—is shown in Figure 7 and Table 3. In Slovenian strains, the most frequently changed amino acids occurred within the motif TSQSTVLDT in the original (amino acids 246–254) and repeated (amino acids 266–274) sequences. The most common mutation in the original region in 69/92 (75%) HRSV-B Slovenian strains was at the second and last positions in the motif TSQSTVLDT, where the amino acid serine (S) was changed with proline (P) and threonine (T) with isoleucine (I) into the motif TPQSTVLDI. Only one change at the second position S to P occurred in 16/92 (17.4%) HRSV-B strains (the motif TPQSTVLDT). The same mutations were not observed in the repeated region. In the original region, a very common mutation was T254I, observed in 76/92 (82.6%) strains. Interestingly, an analogue mutation in the repetitive region was not detected in any of the sequenced samples. The most common motif in the repeated region was TSQSIALDT (83/92; 90.2%). Other motifs in the original (SPQSTVLDI, TPQPTVLDI, TPQSTALDI, TPQSTELDI, and TLQSTVLDI) and repeated regions (TSQPIALDT, TSQSIVLDT, TSQSTALDT, TPQSTALDT, and TSQSIAPDT) occurred less often (Figure 7).

## 4. Discussion

Human RSV, despite SARS-CoV-2, remains one of the most important causative agents of severe respiratory tract infections, especially in young children and infants. Before the SARS-CoV-2 pandemic, HRSV circulated in Slovenia in the late autumn and winter, mostly from October to April, with an epidemic peak in January [8].

To the best of our knowledge, this study is the first comparison of the genetic characteristics of HRSV before, during, and nearing the end of the SARS-CoV-2 pandemic, which also places Slovenian HRSV isolates on the world map for the first time. Phylogenetic analysis of the hypervariable glycoprotein G gene shows two separate clusters, one belonging to subgroup A and another to subgroup B. In 2018 and 2019, HRSV-B dominated over HRSV-A in Slovenian hospitalized pediatric patients, whereas in 2020, before the emergence of SARS-CoV-2, both subgroups were equally distributed. However, the usual respiratory viruses season was interrupted by the emergence and rapid pandemic spread of SARS-CoV-2, in particular by introduced nonpharmaceutical preventive measures. In Slovenia, preventive measures were strictly followed, and between July 2020 and May 2021, HRSV was not detected at all (Figure 1). After 1 year, without detected HRSV, HRSV-positive cases began to reappear in late spring and summer, with the epidemic peak in September 2021. Interestingly, this included the predominance of HRSV-A—which, however, seems in line with previous observations of HRSV subgroup changes between seasons [9,10]. Our results are in concordance with other studies, observing the disappearance of HRSV from the population for the same period of time [14,15,16]. Before the emergence of SARS-CoV-2, two genetic lineages of HRSV circulated in Slovenia (GA2.3.5 and GB5.0.5a). Surprisingly, the same two lineages also remained present after the preventive measures were eased, but in different ratios (Table 1). This indicates that, although HRSV was clearly hampered by nonpharmaceutical measures against SARS-CoV-2, it must have survived in the population and was probably overlooked by a significant drop in targeted testing (more than 118 thousand samples were sent only for the detection of SARS-CoV-2). From these results, it does not seem that only a more resilient strain survived and later spread through the population, causing the unusual epidemiologic picture but the prepandemic usual seasonal change in the dominant HRSV-A to B e.g., B to A type happened anyway. We speculate that the unusual epidemiological pattern is, therefore, most probably linked to human behaviour (longing for interaction) than a specific genetic lineage of the virus that makes it more resistant to environmental factors. On the other hand, Eden et al. suggested that one of the proposed explanations for such novel epidemiology was a reduced genomic diversity in the post-COVID-19 period [17] which, however, does not seem to be the case in our study population. Therefore, it would seem that HRSV incidence surges could be more complex and occurred probably due to the combined action of several factors: alleviation of SARS-CoV-2 prevention measures, reduced genomic diversity, the rebound of respiratory viruses as soon as the other one decreased its spreading, and, finally, better fitness of certain lineages that allow them to survive in the environment [17,18].

All Slovenian HRSV-A strains contained the 72-nucleotide-long duplicated region in the attachment glycoprotein G gene, closely related to the original Canadian ON1 strain, and were additionally classified as lineage GA2.3.5. All Slovenian HRSV-B strains similarly contained the 60-nucleotide-long duplicated region in the attachment glycoprotein G gene, closely related to the English BA, and were additionally classified as lineage GB5.0.5a. These two genotypes also currently dominate the world map of known HRSV lineages (https://doi.org/10.21105/joss.03773, accessed on 28 February 2023), confirming that our population is no exception to the worldwide evolution pattern of HRSV. Amino acid-mutation analysis of the hypervariable attachment protein of the HRSV-A and HRSV-B subgroups focused on the original and duplicated, or repeated, regions (Figure 6 and Figure 7). Within the HRSV-A subgroup, lineage GA2.3.5 shows numerous amino acid changes that seem to occur independently in the original and repeated region, and also with variable frequencies in both regions (Figure 6). In the highly conserved sequence motif GYLSPSQ in both regions, seven different motifs (GYLSPSQ, GYPSPSQ, GYLNPSQ, GYPSLSQ, GYPSQSQ, GYPGPSQ, and GHPSPSQ) were found in the original region and eleven (GYLSPSQ, GYPSPSQ, GHPSPSQ, GYPSPSK, GYPSSSQ, DYLSPSP, GHLSPSQ, GYLSSSQ, SYPSPSQ, GHPGPSQ, and SHPSPSQ) in the repeated region. This is in contrast to HRSV-B, where 90.2% of strains had the same mutation in the repeated region. The most common motif in HRSV-A strains, GYPSPSQ, appeared only in 20.9% (27/129) and 36.4% (47/129) in the original and repeated regions, respectively. All other motifs in both regions occurred at lower frequencies. The motifs GYPSPSQ and GYLSPSQ were already observed previously in Mexican hospitalized children [19]. In the Slovenian HRSV-B subgroup, seven different motifs (TPQSTVLDI, TPQSTVLDT, SPQSTVLDI, TLQSTVLDI, TPQPTVLDI, TPQSTALDI, and TPQSTELDI) in the original region and six (TSQSIALDT, TSQPIALDT, TSQSIVLDT, TSQSTALDT, TSQSIAPDT, and TPQSTALDT) in the repeated region were observed, with the most common motif being TPQSTVLDI (69/92, 75%) in the original region with two mutations and TSQSIALDT (83/92, 90.2%) in the repeated region with two mutations (Figure 7). After the reduction of preventive/protective measures, no new amino acid motifs in the observed region of the HRSV-B subgroup were detected, whereas, in the HRSV-A subgroup, three new motifs (GYPSQSQ, GYPGPSQ, and GHPSPSQ) were observed in the original region and three (SYPSPSQ, GHPGPSQ, and SHPSPSQ) in the repeated region. All amino acid substitutions within the HRSV-B subgroup have already been demonstrated by Kamau et al. [20] except two: one at position 251 (E instead of V or A) and the second at position 258 (E instead of K, I, or Q). Based on these results, Slovenian HRSV-A strains seem to be more diverse than HRSV-B strains. The fact that no circulation of HRSV variants was detected in the population for 1 year (from July 2020 until May 2021), but then there was an epidemic in the middle of the summer of 2021, is quite an unusual HRSV pattern. We expected a specific genetic lineage to be responsible for this observation; however, it does not seem that this is the case.

On the other hand, perhaps this is precisely why these differences are greater within subgroup A than in B because the latter did not circulate with a high prevalence after the emergence of SARS-CoV-2. Based on the fact that no significant differences have been observed within lineages that circulated before and after the SARS-CoV-2 epidemic, coupled with the fact that the same lineages were observed in both periods and the fact that we observed a drop in samples sent for the detection of respiratory viruses after the emergence of SARS-CoV-2, we can conclude that HRSV probably did not disappear completely from the population, but most likely we simply did not detect it. Furthermore, the study population was hospitalized children. All other children that did not have a difficult clinical progression of the disease were probably tested only for SARS-CoV-2 as outpatients and the median age of hospitalized children might have changed during the pandemic. Therefore, we believe that HRSV might have circulated in the population, but with only a mild clinical presentation, and that it was overlooked due to testing focused solely on SARS-CoV-2.

The main limitation of this study is a sample inclusion bias because only samples from hospitalized children were included and the huge amount of samples sent for solely SARS-CoV-2 testing during the time of the study were left out. Moreover, only the differences in short sequences of the second hypervariable glycoprotein region, which is responsible for distinguishing between different lineages, were subjected to sequencing analysis.

## 5. Conclusions

In conclusion, during the 3-year period of the study, only two HRSV genetic lineages circulated within Slovenian pediatric patients: GA2.3.5 (HRSV-A) and GB5.0.5a (HRSV-B). No significant differences were found within strains detected before the SARS-CoV-2 pandemic, during it, and after the implementation of nonpharmaceutical preventive measures. Based on the results of this study, it has been observed that mutations can occur independently inside and outside the hypervariable region, especially within the HRSV-A subgroup. Therefore, it would be of great interest in the future to continue this survey with whole-genome sequencing and further analysis. Finally, it would also be of great importance to monitor the long-term impact of endemic SARS-CoV-2 circulation in the future population on the formation of new HRSV lineages and epidemiological patterns.

## Figures and Tables

**Figure 1 viruses-15-01218-f001:**
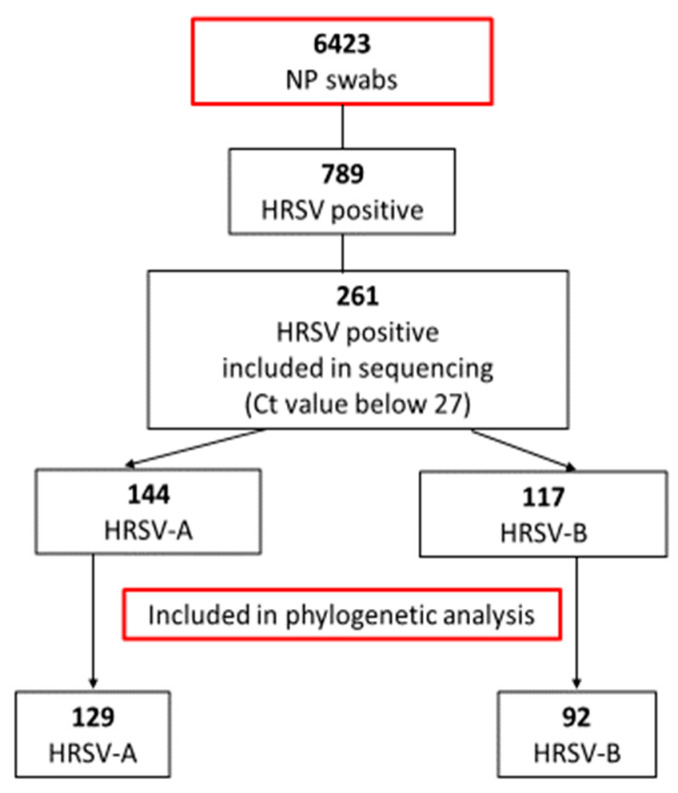
Flowchart of specimens included in the study. NP = nasopharyngeal.

**Figure 2 viruses-15-01218-f002:**
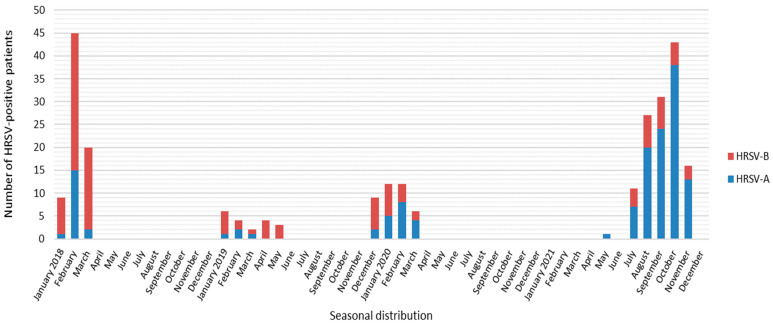
Seasonal distribution of HRSV-positive samples in Slovenia during 2018 and 2021. Blue bars represent the HRSV-A genotype and red the HRSV-B genotype.

**Figure 3 viruses-15-01218-f003:**
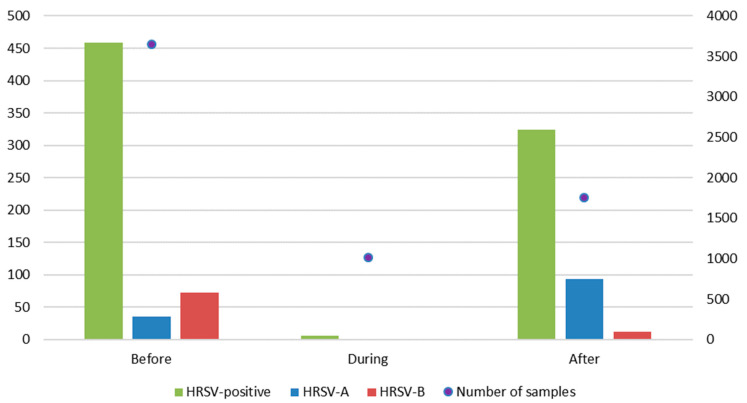
Distribution of total samples for respiratory viruses’ molecular diagnostics in correlation with detected HRSV cases among pediatric patients before the SARS-CoV-2 pandemic from January 2018 to March 2020 (Before) and introduction of preventive measures from March 2020 to March 2021 (During) and after measures started being lifted from March 2021 to December 2021 (After). The left axis shows the number of HRSV-positive samples with red bars representing the total number of HRSV-positive samples, green HRSV-A, and purple HRSV-B. The right axis shows the total number of samples tested for respiratory viruses (blue dots).

**Figure 4 viruses-15-01218-f004:**
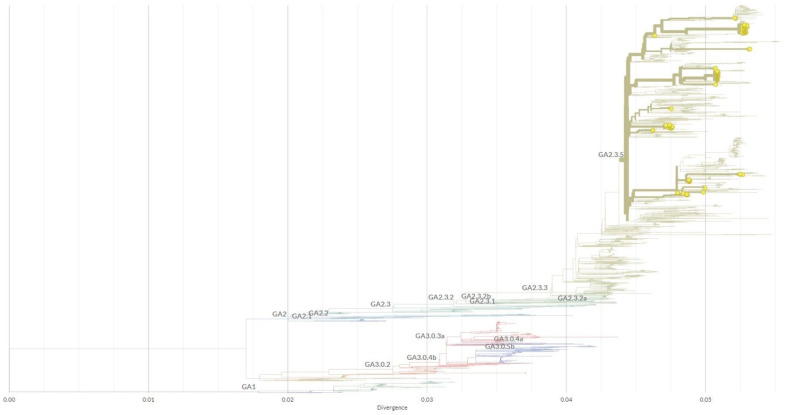
Nextclade [13] phylogenetic tree and genotype determination based on 357 nucleotide attachment glycoprotein G gene sequences of 129 Slovenian HRSV-A cases (GeneBank acc. nos. OQ343511–OQ343639) and the original Canadian ON1 genotype (GenBank acc. no. JN257693).

**Figure 5 viruses-15-01218-f005:**
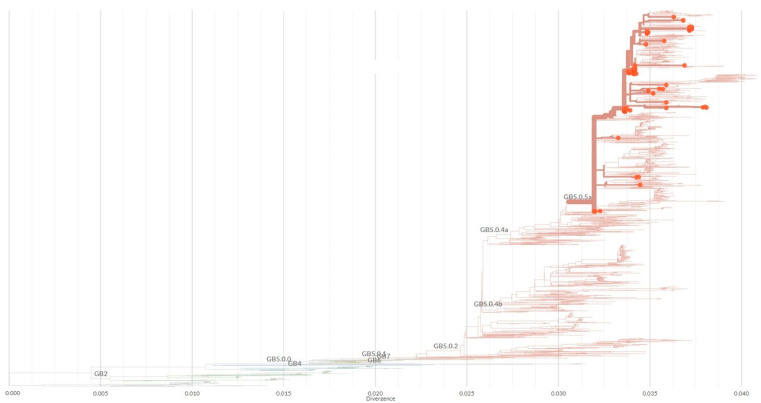
Nextclade [13] phylogenetic tree and genotype determination based on 362 nucleotide attachment glycoprotein G gene sequences of 92 Slovenian HRSV-B cases (GeneBank acc. nos. OQ434465–OQ434556) and the original Canadian ON1 genotype (GenBank acc. no. KY249660).

**Figure 6 viruses-15-01218-f006:**
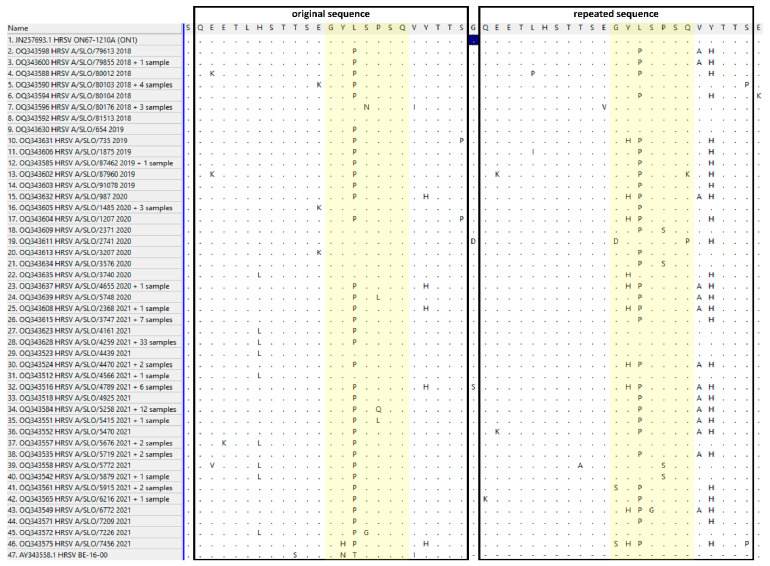
Amino acid changes in the 24-amino acid (72-nucleotide)-long duplicated hypervariable region of the attachment glycoprotein G gene of the HRSV-A subgroup for the 129 Slovenian strains ordered by date of collection in comparison to the original Canadian ON1 strain (GenBank acc. no. JN257693) and a strain without insertion (GenBank acc. no. AY343558). Yellow color indicates highly conserved sequence motif in original and repeated region.

**Figure 7 viruses-15-01218-f007:**
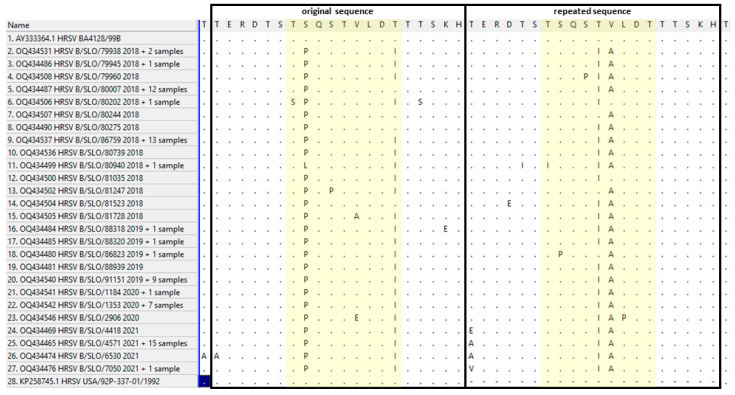
Amino acid changes in the 20-amino acid (60-nucleotide)-long duplicated hypervariable region of the attachment glycoprotein G gene of the HRSV-B subgroup for the 92 Slovenian strains ordered by date of collection in comparison to the original Madrid BA strain (GenBank acc. no. AY333364) and a strain without insertion (GenBank acc. no. KP258745). Yellow color indicates highly conserved sequence motif in original and repeated region.

**Table 1 viruses-15-01218-t001:** Yearly distribution of NP swabs from pediatric samples tested for respiratory viruses, number of HRSV positives, determined subgroup, and genotype.

Year	Total Samples	HRSV Positive [%]	No. Genotyped Samples [%]	Subgroup	No. Samples	Genetic Lineage
2018	1791	262 [14.6]	59 [22.5]	A	15	GA2.3.5
				B	44	GB5.0.5a
2019	1316	116 [8.8]	24 [20.7]	A	7	GA2.3.5
				B	17	GB5.0.5a
2020	1241	86 [6.9]	25 [29.1]	A	14	GA2.3.5
				B	11	GB5.0.5a
2021	2075	325 [15.7]	113 [34.8]	A	93	GA2.3.5
				B	20	GB5.0.5a
Total	6423	789 [12.3]	221 [28.0]	A	129	GA2.3.5
				B	92	GB5.0.5a

**Table 2 viruses-15-01218-t002:** Detailed description of detected amino acid mutations in the repetitive region of HRSV-A gene G relative to genotype ON1.

AA Position ^†^	Relative Position in Rep. ^‡^ Region	AA ON1	Mut. *	No. [%] Samples	AA Position ^†^	AA ON1	Mut. *	No. [%] Samples
261	1	Q	N/A		285	Q	K	2 [2.3]
262	2	E	K	3 [2.3]	286	E	K	2 [2.3]
			V	1 [0.8]				
263	3	E	K	3 [2.3]	287	E	N/A	
264	4	T	N/A		288	T	N/A	
265	5	L	N/A		289	L	P	1 [0.8]
							I	1 [0.8]
266	6	H	L	46 [35.6]	290	H	N/A	
267	7	S	N/A		291	S	N/A	
268	8	T	N/A		292	T	N/A	
269	9	T	N/A		293	T	A	1 [0.8]
270	10	S	N/A		294	S	N/A	
271	11	E	K	10 [7.5]	295	E	V	4 [3.1]
272	12	G	N/A		296	G	D	1 [0.8]
							S	2 [2.3]
273	13	Y	H	1 [0.8)	297	Y	H	20 [15.5]
274	14	L	P	112 [86.8]	298	L	P	71 [55.0]
275	15	S	G	1 [0.8]	299	S	G	1 [0.8]
			N	4 [3.1]				
276	16	P	Q	13 [10.1]	300	P	S	5 [3.9]
			L	3 [2.3]				
277	17	S	N/A		301	S	N/A	
278	18	Q	N/A		302	Q	P	1 [0.8]
							K	1 [0.8]
279	19	V	I	4 [3.1]	303	V	A	40 [31.0]
280	20	Y	H	13 [10.1]	304	Y	H	66 [51.2]
281	21	T	N/A		305	T	N/A	
282	22	T	N/A		306	T	N/A	
283	23	S	P	2 [1.5]	307	S	P	6 [4.6]

† relative to JN257693; ‡ repetitive; * mutation.

**Table 3 viruses-15-01218-t003:** Detailed description of detected amino acid mutations in the repetitive region of HRSV-B gene G relative to genotype BA.

AA Position ^†^	Relative Position in Rep. ^‡^ Region	AA BA	Mut. *	No. [%] Samples	AA Position ^†^	AA BA	Mut. *	No. [%] Samples
240	1	T	A	1 [1.1]	260	T	A	17 [18.5]
							V	2 [2.2]
							E	1 [1.1]
241	2	E	N/A		261	E	N/A	
242	3	R	N/A		262	R	N/A	
243	4	D	N/A		263	D	E	1 [1.1]
244	5	T	N/A		264	T	I	2 [2.2]
245	6	S	N/A		265	S	N/A	
246	7	T	S	2 [2.2]	266	T	I	2 [2.2]
247	8	S	P	92 [100]	267	S	N/A	
248	9	Q	N/A		268	Q	N/A	
249	10	S	P	1 [1.1]	269	S	P	1 [1.1]
250	11	T	N/A		270	T	I	88 [95.6]
251	12	V	A	1 [1.1]	271	V	A	89 [96.7]
			E	1 [1.1]				
252	13	L	N/A		272	L	P	1 [1.1]
253	14	D	N/A		273	D	N/A	
254	15	T	I	76 [82.6]	274	T	N/A	
255	16	T	N/A		275	T	N/A	
256	17	T	S	2 [2.2]	276	T	N/A	
257	18	S	N/A		277	S	N/A	
258	19	K	E	2 [2.2]	278	K	N/A	
259	20	H	N/A		279	H	N/A	

† relative to AY33364; ‡ repetitive; * mutation.

## Data Availability

Not applicable.

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
