# Peer review of "Changes in HRSV Epidemiology but Not Circulating Variants in Hospitalized Children due to the Emergence of SARS-CoV-2"

_viruses, 2023, doi:10.3390/v15061218_

Round 1

Reviewer 1 Report

The paper entitled “Changes in HRSV epidemiology but not circulating 2 variants in hospitalized children due to the emergence of 3 SARS-CoV-2” offers new insights into comprehending the changed epidemiology of RSV circulation during and after the COVID-19 pandemic. It presents interesting findings, that should be discussed in comparison with previous works, by adding a few references. 

Major points:

I think the results of this work are crucial in understanding the unseasonal outbreak of HRSV-bronchiolitis seen in the past two years. One of the proposed explanations for such novel epidemiology was a reduced genomic diversity in the post-COVID-19 period, as suggested by Eden et al in Australians (Eden JS, Sikazwe C, Xie R, et al. Off-season RSV epidemics in Australia after easing of COVID-19 restrictions. Nat Commun. 2022;13(1):2884), but it does not seem so according to your reports.

Furthermore, I suggest the surges of HRSV circulation could not be simply due to the alleviation of SARS-CoV2 prevention measures, as being equal the health policies, a turnover of HRSV and SARS-CoV2 waves has been seen starting from the 2021 summer, with one pathogen surging as soon as the other one decreased its spreading, as suggested by Rabbone et al (Rabbone I, Monzani A, Scaramuzza AE, Cavalli C. See-sawing COVID-19 and RSV bronchiolitis in children under 2 years of age in Northern Italy. Acta Paediatr. 2022;111(11):2174-2175.). Please, comment on these points in the discussion.

Minor points:

There are discrepancies between the numbers reported in the flowchart in Figure 1 and those reported in the results session. Please correct them.

Only minor linguistic revision is suggested. 

Reviewer 2 Report

Summary

This was a study about the RSV genotypes in circulation before, during and near the end of the SARS-CoV-2 pandemic in young children, and the impact of that pandemic on RSV circulation and epidemiology. The authors observed that, from 2012-2020), HRSV displayed a regular circulation y pattern (winter), but after March 2020 when SARS was confirmed in Slovenia, that epidemiological patter disappeared for about one year. Further, once the government lifted restrictions, an unusually high number of RSV cases began to appear in summer, despite high temperatures. They sought to understand the impact of the pandemic on HRSV epidemiology. To do so, investigators conducted phylogenetic analysis of the glycoprotein G gene on approximately 85% of HRSV+ samples, revealing two clusters: one to RSV-A and a smaller one to RSV-B. They report that the Slovenian HRSV-A strains contained the 72-nucleotide long duplicated region in the attachment glycoprotein G gene (GA2.3.5) and all Slovenian HRSV-B strains contained a 60-nucleotide long duplicated region in that gene (GB5.0.5a). From 2018-2021 of the study period, no significant differences were observed within strains detected throughout the pandemic. They report that the HRSV-A strains appear to be more diverse than the HRSV-B strains. They suggest that whole-genome investigations would be needed for better monitoring of the longterm impact of SARS-CoV-2 endemic circulation and their effect on new HRSV epidemiology. 

General comments

The authors state that they collected 6423 NPs from hospitalised children with ARI, however they provide no information on the total number of such children. Was this the total set of eligibles or were there eligible children who were not sampled? It would be important to know the full set, in addition to any sampling strategies (eg specific hours, days, a sampling frame) that might have affected which children were sampled, in order to appreciate the potential for bias. This should be considered standard practice in facility-based studies. The sample collection and processing methods appear to be straightforward and standard, including the testing for other common respiratory viruses.  Apart from having ARI (or ARTI), for which the authors should provide a definition for comparison to other studies, they should also provide other inclusion criteria, such as age-groups, as most such studies tend to focus on younger, more vulnerable children < 59 months. No inclusion criteria are provided. And as the authors note, the median age of infected children was lower than the median age of the paediatric hospitalised population. This too might explain the somewhat lower than expected fraction of RSV-infected children, as the median age of hospitalised children might have changed during the pandemic. 

Were the 221 samples used for further genetic analysis the only ones of suitable sequence quality, or were there any other criteria used to select these? 

The authors speculate in the discussion that the same two lineages remained after the preventive measures were eased, but in different ratios. While they suggest that this might be more the effect of human behaviour, what do they think of the flip in lineage ratios? If this is not due to environmental or innate viral fitness factors (whatever these might be), how do the authors account for the change in ratio? This calls to mind the change in influenza A (H1N1) from its previous seasonal variant to the pandemic variant that never reverted post-influenza pandemic in 2009, suggesting superior viral fitness. Can they think of a way in which this could be tested and how might this affect their further surveillance going forward? 

The authors’ speculation that, because no significant differences were observed within lineages circulating before and after the SARS pandemic, HRSV did not disappear from the population, but was simply not detected (likely due to sampling methodology ie hospitalised patients) is highly probable. Going back to the previous question, how might this affect their surveillance strategy in future? Would random sampling of community controls add value, as has been observed in other longitudinal and case control studies?  

The authors’ statement on limitations (bias) is sound, but difficult to assess in being complete in the absence of information about the selected patients (see comments above). Hence the need for more information regarding inclusion criteria and the number (proportion) of included vs non-included patients. 

Specific comments 

None. 
